# Subjective length of life of European individuals at older ages: Temporal and gender distinctions

Dimiter Philipov[1][☯], Sergei Scherbov[1,2,3][☯] *

1 International Institute for Applied Systems Analysis, World Population Program, Wittgenstein Centre for Demography and Global Human Capital (Univ. Vienna, IIASA, VID/ÖAW), Laxenburg, Austria, 2 International Laboratory on Demography and Human Capital, Russian Presidential Academy of National Economy and Public Administration, Prospekt Vernadskogo, Moscow, Russian Federation, 3 Vienna Institute of Demography, Austrian Academy of Science, Vienna, Austria

☯ These authors contributed equally to this work.
* scherbov@iiasa.ac.at

**Data Availability Statement:** This paper uses data from the Human Mortality Database (18), as well as SHARE Waves 1 and 6 (DOIs: 10.6103/SHARE.w1.700, 10.6103/SHARE.w6.700, see (21) for methodological details.

## Abstract

This paper examines how older individuals living in 9 European countries evaluate their chances of survival. We use survey data for the years 2004 and 2015 to construct population-level gender-specific subjective length of life (or subjective life expectancy) in people between 60 and 90 years of age. Using a specially designed statistical approach based on survival analysis, we compare people's estimated subjective life expectancies with those actually observed. We find subjective life expectancies to be lower than actual life expectancies for both genders in 2004. In 2015 men become more realistic in the sense that their subjective life expectancy is close to what was actually observed, while women retain their subjective expectations of a shorter than actual life expectancy. These results help to better understand how people might construct diverse decisions related to their remaining life course.

## Introduction

Older individuals construct expectations about their remaining length of life with respect to where they live, their health, their life style, their family history, and other factors. They organize their remaining life course and take a wide range of crucial decisions based on these expectations, for example: distribution of savings and properties; changes in employment status and retirement; family issues and living arrangements; bequests; personal health; and care provision. Personal expectations about remaining life years are innately probabilistic because individuals are unaware of what their actual longevity will be. Scientists explore these expectations with survey data which they use to estimate diverse indicators of survival.

An important issue is the extent to which subjective measures of survival reflect actual survival. A mismatch between subjective expectations and the actual length of remaining life can lead to biased decision-making related to the remaining life course. There may be undesired consequences if planned life course episodes and events occur at the "wrong" time, which can

**Funding:** This project has received funding from the European Union's Horizon 2020 research and innovation program under grant agreement No 635316 (Project Name: Ageing Trajectories of Health: Longitudinal Opportunities and Synergies, ATHLOS), SS. The SHARE data collection has been funded by the European Commission through FP5 (QLK6-CT-2001-00360), FP6 (SHARE-I3: RII-CT-2006-062193, COMPARE: CIT5-CT-2005-028857, SHARELIFE: CIT4-CT-2006-028812), FP7 (SHARE-PREP: GA N°211909, SHARE-LEAP: GA N°227822, SHARE M4: GA N°261982) and Horizon 2020 (SHARE-DEV3: GA N°676536, SERISS: GA N° 654221) and by DG Employment, Social Affairs & Inclusion. Additional funding from the German Ministry of Education and Research, the Max Planck Society for the Advancement of Science, the U.S. National Institute on Aging (U01_AG09740-13S2, P01_AG005842, P01_AG08291, P30_AG12815, R21_AG025169, Y1-AG-4553-01, IAG_BSR06-11, OGHA_04-064, HHSN271201300071C). The funders had no role in study design, data collection and analysis, decision to publish, or preparation of the manuscript.

**Competing interests:** The authors have declared that no competing interests exist.

increase frustration, anxiety, and stress. Such issues demonstrate the need for analyses of potential deviations of subjective measures of longevity from their objective equivalents. This topic has received less attention than analyses of subjective survival, particularly with regard to the elderly population.

Path-breaking research [1] based on survey data compares subjective survival probabilities and corresponding life expectancies for males in the United States with those actually observed at the time of survey. The findings show that subjective life expectancies are slightly higher than actual ones. Comparisons of subjective with actual survival plummeted when data from the Health and Retirement Survey (HRS) started being used. One study [2] reports that survey measures of subjective survival probabilities aggregate well to the corresponding population probabilities; another study [3] based on four waves of HRS data reports that subjective longevity expectations are consistent with individuals' observed survival patterns. A comparison of subjective and actual probabilities using longitudinal HRS data arrived at similar conclusions [4]. However, differences in hazards corresponding to subjective and actual survival curves estimated based on longitudinal HRS data show that both men and women overestimate their remaining life expectancy [5].

Several studies discuss this validation for the elderly in Europe using data from Survey of Health, Ageing and Retirement in Europe (SHARE). Above the age of 60, both men and women overestimate their subjective survival probabilities, with men being less accurate [6]; analysis for 11 countries based on a longitudinal dimension of 2 years (the period between Wave 1 and Wave 2 of SHARE) showed that subjective life expectancy corresponds to actual life expectancy" [7]; a study of 9 countries concluded that male subjective survival probabilities are close to the probabilities computed from cohort life tables, whereas female subjective probabilities are lower [8].

Other studies based on different data sets report different conclusions. Analysis of longitudinal data from the Berlin Ageing Study collected over 16 years shows that individuals aged 70 and over have relatively accurate perceptions of their nearness to death [9]. An exploration of the German SAVE panel (*Sparen und Altersvorsorge in Deutschland*) over the 25–60 age span inferred that men and women underestimate their longevity in a cohort perspective by 6–7 years [10]. Survey data for France came to similar conclusions [11]; in this survey each respondent stated his or her chances of survival to several different ages, which enabled individual survival curves and relevant individual life expectancies to be estimated. The findings show that respondents over 60 underestimate their subjective life expectancy relative to life table life expectancy, and that this gap is larger for women. The two studies show that lower subjective life expectancies relative to actual life expectancy are observed for both genders.

This brief outline shows that while studies based on HRS data for the United States report a satisfactory match between subjective and actual survival rates, studies in Europe are not as unified; some report a good match, while others show that respondents, and particularly women, underestimate their potential longevity. Differences in data, measurement, and methods of estimation are the main reasons for diversity in the findings. Some surveys are longitudinal and provide individual-level data suitable for comparisons of an individual's expectation of survival to a certain age with the actual survival of the same individual. Other surveys are cross-sectional, where the integration of data provides survival rates related to the study of a population. Survey questions also differ and may refer to the measurement of different indicators of longevity. Accordingly, methods of research should comply with each specific data set and measurement. This overall diversity hinders comparisons of findings, as each finding is valid within its own scope of data, measurement, and method.

This paper contributes to research based on the use of subjective life expectancies. Although these are derived from survival probabilities, we prefer them because they serve as an

integrated indicator of longevity that is easy to understand, easy to communicate, and easy to use for comparisons across time, space, and gender. Research on the health status of the elderly supports our choice, as healthy and unhealthy life expectancies estimated using respondents' subjective views of their health are routinely used as indicators [12,13].

These remarks are intended to raise awareness of the following topics of research which we consider to be insufficiently covered in the literature:

1. International comparisons can help to ascertain whether deviations are systematic or not.

2. Temporal analyses may indicate whether a fit remains stable in time or whether a misfit increases or decreases with time.

3. Differences between the two genders need to be the subject of further deliberations.

We deal with these research topics at the macro level with a focus on the over-60 population. We develop a method to estimate population-level subjective life expectancies based on observed individual subjective survival probabilities and compare them with actually observed life expectancies. The next section describes data and methods. It is followed by a description of estimated results; a discussion of these can be found in the final section.

## Data and methods

We use SHARE data from Wave 1 (2004) and Wave 6 (2015) from which we selected men and women aged 60 to 80 years. The upper age boundary was chosen because institutionalization increases in the over-80 population. Subjective survival probabilities are observed with responses to one question: "What are the chances that you will live to be age T or more?" The target age T depends on the age of the respondent as indicated in Table 1.

As this question may seem difficult to respondents, they were introduced to the issue of "chances" with an example. They were asked to answer the question: "What do you think are the chances that it will be sunny tomorrow?" with answers from 0 to 100.

We selected 9 countries where data were available in both waves: Austria, Belgium (Wave 1 in 2005), France, Germany, Greece, Italy, Sweden, Spain, and Switzerland.

The number of "no response" to the question about survival chances was low and is not expected to influence our results. However, the answer "don't know" to the same question accounted for about 9 percent of all respondents eligible for our study in Wave 1 and for 7 percent in Wave 6. The proportion of "don't knows" is higher in France, Greece, Italy, and Spain in both waves. We compared these answers to those about limited abilities versus non-limited abilities during the 6 months prior to the survey, measured with the Global Activity Limitation Indicator (GALI) [12–14]. The answer "don't know" to chances of survival was prevalent among respondents who declared having limited abilities. We assume that people with limited abilities will state lower survival probabilities compared to those with non-limited abilities; hence, if respondents had answered with their chance of survival, instead of selecting "don't know," survival at the population level would decline.

Table 1. Target ages (T) in the SHARE question about chances for survival to age T.

| Age of the respondent | Target age T |
|---|---|
| 60 to 65 | 75 |
| 66 to 70 | 80 |
| 71 to 75 | 85 |
| 76 to 80 | 90 |

Answers to the question could be any number between 0 and 100; 0 reflects the respondent's expectation that he/she will certainly not live to the target age, and a value of 100 reflects full certainty of survival to this age. A significant proportion of the answers are clustered around some focal points, with the majority at value 50 (around 26 percent of our selection of respondents); this answer reflects "epistemic uncertainty" [15]. Another large focal point is 100 (around 15 percent). In contrast to other methods of estimation, focal points do not constitute important problems for the method of estimation used in this paper.

When the answer is divided by 100, the question measures the probability $S_{x,i}(T)$ of an individual i aged x years at the time of the survey surviving to age T or higher. By definition, it is analogous to the corresponding life table function, but differs in two respects: the probability $S_{x,i}(T)$ relates to an individual and reflects subjective views expressed at the time of the survey; the life table probability refers to a population and reflects the probability of actual survival. We assume that a population defined by a specified country, gender, and year pertains to one life table based on actual observations. Pulling together values of individual $S_{x,i}(T)$ leads to a subjective life table of the same study population. We aim to compare life expectancies estimated in actual and subjective life tables. These comparisons are used for analyses of our research topics.

The issue is how to pull together values of the subjective $S_{x,i}(T)$ into a survival function of one life table. A preferred approach reported in the literature is to consider each response as a draw from a particular distribution that describes the corresponding population's life table survival function. Gompertz and Weibull distributions are most frequently used in this regard (for example [5] and [16]). Other distributions are also applied, for example, the beta distribution [17] and the logistic distribution [4]. Application of statistical methods such as non-linear programming or maximum likelihood can provide estimates of the parameters of these distributions. Preferences point to the Gompertz function with its long-time applications for the study of mortality. The Gompertz function for survival from age 0 to age x is:

$$S_0(x) = \exp\left({}^{\alpha}/_{\beta}.(1 - \exp(\beta.x))\right)$$

with parameters α and β. The following life table equality links the observed survival probabilities $S_x(T)$ with $S_0(x)$: $S_x(T) = S_0(T)/S_0(x)$. The estimated survival probabilities can be used to compute life expectancies.

**Table 2. Algorithm of the model applied for the estimation of subjective life expectancies.**

| | |
|---|---|
| Step 1 | For each respondent i, one number $J_i$ is drawn from the uniform distribution [0,1]. |
| Step 2 | Each respondent's subjective $S_{x,i}(T)$ is compared with this number. If $S_{x,i}(T) \geq J_i$ the person is supposed to have survived to age T; if $S_{x,i}(T) < J_i$, the person is supposed to have died at an unknown age in the interval [X, T). This survival outcome is probabilistic and in line with the probabilistic nature of $S_{x,i}(T)$. |
| Step 3 | A dichotomous variable Z is constructed which denotes survivals to age T as 1 and deaths in the interval (X, T) as 0. |
| Step 4 | Survival analysis is applied for interval-censored data assuming a Gompertz distribution of survival, using a maximum-likelihood estimator for the parameters of the Gompertz function. At this step we applied weights as available in the survey. |
| Step 5 | A life table $S_x(T)$ is estimated using the Gompertz function specified with the received parameters. The life expectancy of 60-year olds is estimated following conventional formulas in a life table, in the 60–90 year age interval. |
| Step 6 | The procedure is repeated from Step 1 to Step 5 to obtain another value for the life expectancy. |

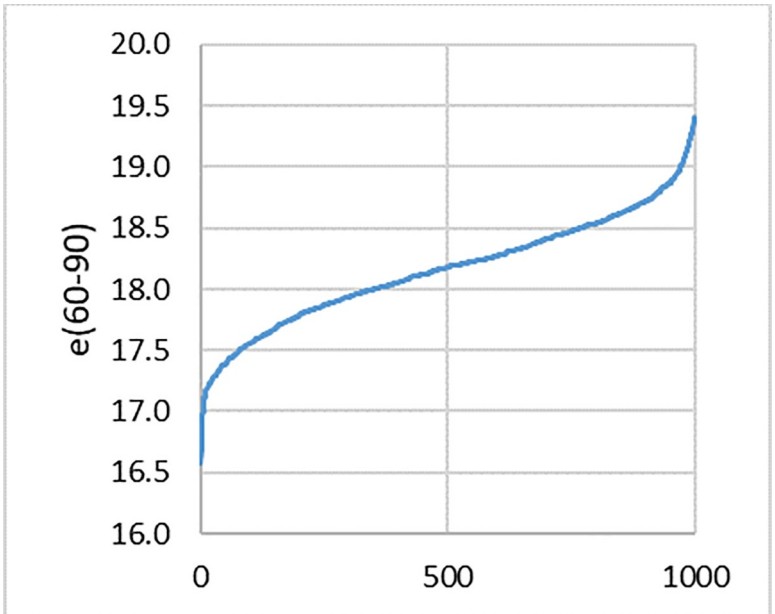

**Fig 1. Empirical distribution of 1000 segmented subjective life expectancies in the age interval 60–90 estimated with 1000 draws, German males, 2004 SHARE Wave 1 data; values ordered from low to high.**

When non-linear regression is applied, confidence intervals of parameter estimates cannot be directly applied to estimating confidence intervals of life expectancies; additional modeling and additional assumptions are required. We prefer a different approach. As the task of pulling together individual survival probabilities belongs to survival analysis, methods developed within this statistical field are preferable. Hence, it is possible to make use of the probabilistic nature of the individuals' responses about their chances of survival because each response is a draw from an unknown subjective probability distribution (a general discussion on subjective probability distributions is available in [18]).

We developed a model approach that comprises the steps included in Table 2.

Our choice of the ages 60–80 for the respondents and 60–90 for the life expectancies needs clarification. We restrict the respondents' upper age to 80 because the proportion of institutionalized persons rises significantly above this age, and they are not included in the survey samples. Estimates of subjective life expectancies refer to the age interval 60–90 because answers about subjective survival to target age 90 were given by respondents aged 76–80. The life expectancies thus estimated are the sum of person-years lived in the age interval between 60 and 90; we refer to them as segmented to differentiate them from the classical equivalents which are not limited by an upper survival age.

In this procedure, the value of the (subjective) life expectancy depends on random numbers $J_i$. Hence, its value is also random and is a draw from an unknown distribution. We construct the latter empirically, repeating 1000 times the execution of steps 1 to 5. Fig 1 gives the form of this distribution for German males, Wave 1 (2004).

If the draws were a different number, such as 500 or 2000, the distribution would display essentially the same shape.

The mean of all 1000 life expectancies is an estimate of the "true" one; for German males in 2004 it is equal to 18.2 years. It is inappropriate to estimate standard errors and construct confidence intervals around the mean based on this empirical distribution, as the sample size, 1000 in this case, depends on the researcher's decision. Instead of working with standard

errors and confidence intervals, we preferred to use 2.5 and 97.5 percent percentiles to define intervals for the estimated mean of the life expectancy similar to the confidence intervals; for German males in 2004 these are 17.4 and 18.9 years.

We compare subjective life expectancy to age 90 with actual life expectancy for the whole population estimated from observed data for the age interval 60–90. The actual life expectancy is thus also segmented. These life expectancies were estimated using data from the Human Mortality Database (HMD) [19] for the years 2004 (Belgium 2005 to fit the year of Wave 1) and 2015 (last available data in Greece were for 2013 and in Italy for 2014).

## Results

This section displays the results from the estimation procedures. They are considered in the discussion in the next section.

We report subjective and observed life expectancies for males and females, segmented over the 60–90 age interval for the years 2004 and 2015 (we skip further the specification "segmented" for life expectancy; all life expectancies in this paper are segmented). We report percentiles of the distribution of estimated subjective life expectancies and use them to elucidate the statistical significance of differences among life expectancies (LE). When an actual LE is outside the range defined by the two percentiles of the corresponding subjective LE, we can reject the hypothesis that the actual and the subjective LE do not differ.

We first display results for the year 2004, then for 2015, followed by the results for temporal change and gender differences.

### Results for 2004

Table 3A shows that for males the average subjective LE for nine countries is lower than the actual one by slightly over one year. At the country level, the highest subjective LE is observed in Switzerland where the actual LE is also highest of the 9 countries. Subjective LE is also high in southern Europe, specifically in Spain, Italy, and Greece but this observation does not hold for the actual LE. Differences between subjective and actual LE are displayed in the far right-hand column of the table, statistically evaluated using percentiles. Numbers in bold indicate where the difference is beyond the percentile interval. It is worth noting that statistically significant values are greater than one year, and differences cannot be accepted as significant when they are less than one year.

We find that in 2004, males in Switzerland and southern Europe had a realistic view about perspectives of their remaining life years. In the remaining five countries—Austria, Belgium, France, Germany, and Sweden—men's views about the length of their life show them to be biased downwards: French men appear more downwardly biased, as their expectations about length of life differ from the actual length of life by two years. We can also ignore confidence intervals because differences displayed in the far right-hand column of the table are not as low and could be substantively meaningful. One dominant feature then emerges: men in all 9 countries underestimate their length of life by at least half of one year, and men in Spain by 0.3 of one year.

Table 3B gives similar data for women. An obvious inference can be made from the results: women in all countries exhibit a downward bias, as the length of life they expect is nearly 5 years lower than their actual lifespan. Downward bias is particularly strong among French women where the subjective LE is nearly 6 years lower than the actual. Women in Austria and Germany also underestimated their length of life.

**Table 3. a) Segmented subjective and actual life expectancy (LE) and percentiles of the subjective LE, males, 9 European countries, 2004.**

| | Subjective LE | | | Actual LE | Actual–Subj. |
|---|---|---|---|---|---|
| | **Mean** | **Percentiles** | | | |
| | | **0.025** | **0.975** | | |
| A | | | | | |
| Austria | 18.7 | 17.5 | 19.9 | 20.1 | *1.4* |
| Belgium | 18.3 | 17.4 | 19.2 | 19.8 | *1.5* |
| Switzerland | 20.5 | 18.7 | 22.1 | 21.3 | 0.8 |
| Germany | 18.2 | 17.3 | 19 | 19.7 | *1.5* |
| Spain | 20.2 | 18.9 | 21.5 | 20.5 | 0.3 |
| France | 18.7 | 17.4 | 19.8 | 20.7 | *2* |
| Greece | 19.7 | 18.5 | 20.9 | 20.3 | 0.6 |
| Italy | 20 | 18.9 | 21.1 | 20.7 | 0.7 |
| Sweden | 19.2 | 18.3 | 20 | 20.8 | *1.6* |
| Average | 19.3 | 18.1 | 20.4 | 20.4 | *1.2* |
| B | | | | | |
| | | 0.025 | 0.975 | | |
| Austria | 18.2 | 17.1 | 19.4 | 23.4 | *5.1* |
| Belgium | 18.6 | 17.6 | 19.5 | 23.3 | *4.7* |
| Switzerland | 20.4 | 18.7 | 21.9 | 24.2 | *3.9* |
| Germany | 18.1 | 17.2 | 19 | 23.2 | *5* |
| Spain | 19.8 | 18.6 | 20.9 | 24.3 | *4.5* |
| France | 18.8 | 17.7 | 19.8 | 24.6 | *5.8* |
| Greece | 18.4 | 17.2 | 19.6 | 23.2 | *4.9* |
| Italy | 19.5 | 18.4 | 20.5 | 24.1 | *4.7* |
| Sweden | 19.6 | 18.8 | 20.5 | 23.5 | *3.9* |
| Average | 19 | 17.9 | 20.1 | 23 | *4.7* |

Source: Authors' estimates based on SHARE data, Wave 1, 2004 (Belgium 2005), and Human Mortality Database. In bold observations where the actual LE is outside the boundaries of the two percentiles. Segmented LE refer to the age interval 60–90.

## Results for 2015

Next, we examine results observed in 2015. These are displayed in Table 4A for males and Table 4B for females.

Compared to 2004, males in Switzerland and southern Europe maintained their realistic expectations in 2015, while men in Belgium and France retained their downward bias. German and Swedish males changed their previous downward bias to a realistic view. Men in Austria changed their downward bias to an upward bias, but this change needs additional study beyond the scope of this paper.

Downward bias continued to have total prevalence among women in 2015. It is most pronounced in southern European countries plus France. On average, women's overall bias is lower by about 1.5 years in 2015 than in 2004, and is considerably decreased in Austria.

## Temporal change

Table 5 shows the results for temporal change. From 2004 to 2015, actual LE increased by about 1.2 years for males and 0.7 years for females. In the same period, subjective LE increased on average by 2.1 years for males and 2.2 years for females which is twice that of actual LE: subjective estimates of LE increased at a higher pace than actual LE.

**Table 4. a) Segmented subjective and actual life expectancy (LE) and percentiles of the subjective LE, males, 9 European countries, 2015.**

| | Subjective LE | | | Actual LE | Actual–Subj. |
|---|---|---|---|---|---|
| | Mean | Percentiles | | | |
| | | 0.025 | 0.975 | | |
| A | | | | | |
| Austria | 22.6 | 21.6 | 23.5 | 21.3 | *-1.3* |
| Belgium | 20 | 19.2 | 20.7 | 21.1 | **1.2** |
| Switzerland | 22.4 | 21.4 | 23.3 | 22.5 | 0.2 |
| Germany | 21.1 | 20.4 | 21.9 | 20.8 | -0.4 |
| Spain | 21.8 | 20.6 | 22.9 | 21.9 | 0.1 |
| France | 20.2 | 19.2 | 21 | 21.9 | *1.7* |
| Greece | 20.6 | 19.7 | 21.3 | 21.2 | 0.6 |
| Italy | 21.7 | 20.9 | 22.6 | 22.2 | 0.4 |
| Sweden | 22.3 | 21.6 | 23 | 22.2 | -0.1 |
| Average | 21.4 | 20.5 | 22.2 | 21.7 | 0.3 |
| B | | | | | |
| | | 0.025 | 0.975 | | |
| Austria | 22.8 | 22 | 23.5 | 24.1 | *1.3* |
| Belgium | 20.5 | 19.8 | 21.2 | 23.9 | *3.4* |
| Switzerland | 22 | 21.2 | 22.8 | 24.9 | *2.9* |
| Germany | 21.3 | 20.5 | 22 | 23.8 | *2.5* |
| Spain | 21.2 | 20.1 | 22.2 | 25.2 | *4* |
| France | 20.6 | 19.7 | 21.5 | 25.2 | *4.6* |
| Greece | 19.6 | 18.8 | 20.4 | 24.2 | *4.6* |
| Italy | 20.6 | 19.8 | 21.3 | 24.9 | *4.2* |
| Sweden | 22.7 | 22 | 23.4 | 24.2 | *1.4* |
| Average | 21.3 | 20.4 | 22 | 24.5 | *3.2* |

Source: Authors' estimates based on SHARE data, Wave 6, 2015, and Human Mortality Database for 2015 (last available HMD data: Italy 2014, Greece 2013). In bold observations where the actual LE is outside the boundaries of the two percentiles. Segmented LE refer to the age interval 60–90.

**Table 5. Temporal change in subjective and actual life expectancy from 2004 to 2015 (values in 2015 minus values in 2004), males and females.**

| | Males | | Females | |
|---|---|---|---|---|
| | Est. | Actual | Est. | Actual |
| Austria | 3.9 | 1.0 | 4.6 | 0.7 |
| Belgium | 1.6 | 1.3 | 1.9 | 0.6 |
| Switzerland | 1.9 | 1.2 | 1.7 | 0.6 |
| Germany | 3.0 | 1.1 | 3.1 | 0.6 |
| Spain | 1.6 | 1.3 | 1.4 | 0.9 |
| France | 1.5 | 1.2 | 1.8 | 0.6 |
| Greece | 0.8 | 0.9 | 1.2 | 1.0 |
| Italy | 1.7 | 1.4 | 1.2 | 0.7 |
| Sweden | 3.1 | 1.3 | 3.1 | 0.7 |
| Average | 2.1 | 1.2 | 2.2 | 0.7 |

Source: Tables 2 and 3

We evaluate the statistical significance of differences in subjective LE as follows. Consider, for example, Greek males. The value for 2015 is 20.6 (Table 4A) and lies outside the interval defined by the two percentiles for 2004, namely, 17.2 and 19.6 (Table 3A). We thus conclude that the difference of 0.8 of one year (Table 5) is statistically significant. The same inference holds for all other differences between subjective LE in Table 5, and for this reason it is not shown in the table.

At the country level a large increase in subjective LE is notable in the two German-speaking countries, Austria and Germany, and also in Sweden, with more than 3 years for both men and women. In the southern European countries and in Belgium, France, and Switzerland, the increase is under two years for both genders. Values of between 2 and 3 years could be noted only for the averages across the 9 countries.

Fig 2 gives a visual presentation of the increase in subjective LE based on the example of German males. The figure displays the two sets of 1000 subjective LE estimated with data from Wave 1 and Wave 6. The increase is around 3 years, and the lowest value estimated with Wave 6 data is higher than the highest value estimated with data from Wave 1.

## Gender comparisons

The last topic in this section concerns gender comparisons. Averages in Table 6 show that actual LE was around 3 years lower for men than for women in 2004 and in 2015. For subjective LE the observation was different, with the averages for men and women being about equal, and the differences not being greater than 0.2 of one year in 2004 and in 2015. Substantively, in the context of our discussion, 3 years is a significant difference, while 0.2 of one year is not. A check for statistical significance as described in the previous sub-sections leads to an analogous conclusion. We conclude that gender differentials did not change much over time.

Some of the differences seen at the country level could be on the edge of substantive significance, for example, the value of 0.5 of one year observed for Austria in 2004. The difference, however, is statistically insignificant.

Statistical significance holds for differences shown in Greece (2004), and in Greece and Italy (2015). In Italy in 2004, and to some extent in Spain in both years, the differences can be accepted as substantively important. In these three southern European countries, there was greater downward bias regarding survival on the part of women than of men in 2004 and in 2015.

In a figure with two graphs for men and women, similar to Fig 1, the two curves would be hard to distinguish, even for Greece where the largest differences were observed. For this reason, the figure has not been included here.

## Discussion

In this paper we advocate the use of subjective life expectancy as an aggregate measure of subjective probabilities of survival. This indicator is familiar from life table theories and it has important advantages vis-à-vis other measures of survival, such as hazards or rates. Life expectancy, when segmented for a closed age interval, is also much easier to understand and use in analyses. We applied it for cross-country analyses as well as for gender and temporal comparisons.

We developed and applied a new method of estimation within the framework of survival analyses. In this framework, we consider the reported subjective probabilities of survival as draws from a probability distribution. We generated 1000 trials for simulated survival, and for each trial we estimated a subjective life expectancy using a Gompertz survival curve. Percentile levels of a distribution based on these trials served as confidence intervals.

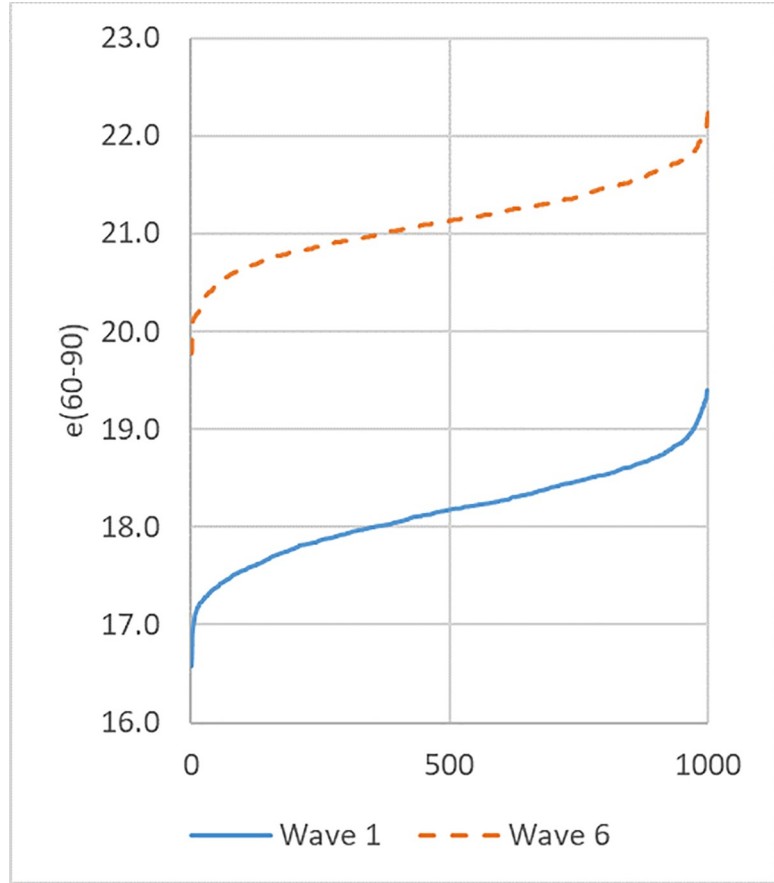

**Fig 2. Subjective LE: 1000 values estimated with data from Wave 1 and 1000 values estimated with data from Wave 6 (exact procedure of estimation described in the previous section).** Values for each wave are ordered from low to high.

We understand subjective probabilities of survival as being personal expectations that reflect a respondent's perception of survival at the time of survey. We consider it unlikely that

**Table 6. Gender differences in estimated and actual life expectancy (men's minus women's), 2004 and 2015.**

|  | 2004 | | 2015 | |
|---|---|---|---|---|
|  | **Est.** | **Actual** | **Est.** | **Actual** |
| Austria | 0.5 | -3.3 | -0.2 | -3.0 |
| Belgium | -0.3 | -3.5 | -0.5 | -2.8 |
| Switzerland | 0.1 | -3.0 | 0.4 | -2.5 |
| Germany | 0.0 | -3.5 | -0.1 | -3.0 |
| Spain | 0.4 | -3.8 | 0.6 | -3.4 |
| France | -0.1 | -3.9 | -0.5 | -3.3 |
| Greece | 1.4 | -2.9 | 0.9 | -3.0 |
| Italy | 0.6 | -3.4 | 1.1 | -2.7 |
| Sweden | -0.4 | -2.6 | -0.4 | -2.0 |
| Average | 0.2 | -3.3 | 0.1 | -2.9 |

Source: Tables 2 and 3

respondents are able to forecast their personal survival for the next 10–15 years. For this reason, we prefer the application of period life tables; moreover, construction of cohort life tables over an age span of 30 years requires the use of assumptions and relevant estimation procedures. Yet, if cohort life tables were considered, then the differences between subjective and actual life expectancy would be even larger because the cohort actual life expectancy would be higher than its period counterpart, in other words, a cohort approach would reinforce our inferences.

We find that in 2004 underestimated views about length of life dominated in all 9 countries studied here. Women underestimate their length of life by nearly 5 years within the age interval 60–90 years when compared with the actual life expectancy, while men's underestimation is lower. These results are consistent with those previously reported for Germany [10] and for France [11], and also with those reported in [8], although these three reports rely on the cohort approach.

During the 11 years between 2004 and 2015, subjective life expectancies increased at a higher pace than actual ones. Gender differences did not change. Underestimation decreased for both genders, and men had become much less unbiased in 2015, unlike women. It is important to see how this tendency will develop in the future. If it continues during the next 5–10 years, men may become upwardly biased while women may closely approach realism. If it does not change much for the men, then they will have reached steady realism about life length. Clearly, though, rigorous conclusions would be crucial as a basis for social and economic policies related to the life course of the elderly.

Specific studies that include the influence of health, education, income and many other personal traits and societal characteristics are required to substantiate our findings. We expect that personal health status will be a central issue with a direct impact on survival expectations. Other issues such as education are also influential in terms of chances for survival, mainly due to their impact on health [20].

There are different reasons why there may be a decline in the downward bias over time. Some are related to the spread of healthy life styles with good diets, an active life, a decline in smoking and alcohol consumption, and other issues related to active aging. Expectations about improved health, and hence about a longer life, are expected to increase. Other reasons would be population composition effects, such as an increase in the proportion of the elderly with higher education who are more meticulous about keeping up a healthy life style. This increase is documented in statistical data at the population level, and it should have been reflected in survey samples (we discuss this issue below). Hence, there is an increase in the proportion of people with better than average health. Improvements in health care are a further reason why there may be a rise in expected subjective length of life. Rigorous conclusions would be valuable for the evaluation and improvement of pertinent social policies.

Our results revealed that men and women have about equal expectations of length of life: around 19 years in 2004 and 21 years in 2015. This result is unexpected because women's actual length of life is longer. Analogous outcomes have been reported for healthy life expectancy measured with subjective statements of survey respondents, for example with GALI.

Numerous studies show that women report significantly worse health than men, and their proportion of healthy life is considerably lower than that of men. This is an unexpected outcome because women live longer and hence are expected to be in better health than men. This has led to what is known as the male–female health-survival paradox [21,22]. It is likely that unhealthy persons will report lower probabilities of survival; hence the pronounced downward bias of women, relative to men, is consistent with their worse health status. Our findings reflect a similar paradox: a male–female subjective survival paradox.

Comparing subjective with healthy life expectancies reveals a crucial advantage for the former. The validity of the comparisons can be tested with evaluations of objectively observed actual life expectancies. Healthy life expectancies do not have an exact population-level statistical observation and hence do not have an anchor for validation. Only indirect validation is feasible, for example, comparing women's health with men's.

The fact that male–female comparisons of healthy life expectancies and subjective life expectancies resemble each other indicates that they are subject to similar explanations. This topic requires further research.

At the country level we find that our main inferences hold in all 9 countries: downward bias for men and women in 2004, downward bias for women in 2015, larger bias for women than for men. We conclude that our inferences are systematic and not a result of some peculiarity. Some country-specific observations require deeper attention. Notably men in the three southern European countries reveal expectations of longevity in 2004 that are close to realistic, and downward bias dominates in France for both genders and in both years.

Our study is not free of caveats. Survey data are usually problematic with respect to how well they represent the population of study, in this case men and women aged 60 to 80 years. The SHARE data have been found to be non-representative regarding educational attainment: persons with higher education are over-represented in our list of countries [23]. Due to this sample bias, estimated subjective life expectancies are higher than those of the study population. If samples were representative for education, subjective life expectancies would be lower and therefore our results would be reinforced, as downward bias would increase. An open question remains with respect to men's realism in 2015; it could turn out that men were biased downwards in 2015 as well. To check the effect of non-representativeness by education we used crudely constructed weights by education for Austrian men in 2015 using their distribution by three levels of education [23,24]. We found that the subjective life of these men would be a fraction of one year lower (i.e., 0.2) than the one displayed in Table 4A. We thus do not expect that non-representativeness by education harms our inferences.

To conclude, we raise some policy implications that refer to the life course of the elderly. Men and women who underestimate their length of life are likely to construct their life course over a span that will be shorter than the actual. Later in life they may be confronted with undesirable situations related to savings, investment, bequests, and living arrangements. Hence, we need more information about the prevalence of this kind underestimation and its effect on introducing bias into life course decisions. The underestimation may induce other negative aspects, for example, increased stress and anxiety. Last but not least, differences between men and women may also indicate that unisex policies might lead to gender inequalities among the elderly.

## Acknowledgments

We would like to thank Stefanie Andruchowitz for her assistance in the preparation of the manuscript.

## Author Contributions

**Conceptualization:** Dimiter Philipov, Sergei Scherbov.

**Formal analysis:** Dimiter Philipov, Sergei Scherbov.

**Investigation:** Dimiter Philipov, Sergei Scherbov.

**Methodology:** Dimiter Philipov, Sergei Scherbov.

**Validation:** Dimiter Philipov, Sergei Scherbov.

**Visualization:** Sergei Scherbov.

**Writing – original draft:** Dimiter Philipov, Sergei Scherbov.

**Writing – review & editing:** Dimiter Philipov, Sergei Scherbov.

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
