## [Decision Letter · Decision Letter 0]

18 Dec 2019

PONE-D-19-30019

Subjective Length of Life of European Individuals at Older Ages: Temporal and Gender Distinctions

PLOS ONE

Dear Dr. Sherbov,

Thank you for submitting your manuscript to PLOS ONE. After careful consideration, we feel that it has merit but does not fully meet PLOS ONE’s publication criteria as it currently stands. Therefore, we invite you to submit a revised version of the manuscript that addresses all the points raised during the review process.

We would appreciate receiving your revised manuscript by Feb 01 2020 11:59PM. To enhance the reproducibility of your results, we recommend that if applicable you deposit your laboratory protocols in protocols.io, where a protocol can be assigned its own identifier (DOI) such that it can be cited independently in the future. For instructions see: http://journals.plos.org/plosone/s/submission-guidelines#loc-laboratory-protocols

We look forward to receiving your revised manuscript.

Kind regards,

Gianluigi Forloni

Academic Editor

PLOS ONE

Journal Requirements:

**When submitting your revision, we need you to address these additional requirements:**

**Please ensure that your manuscript meets PLOS ONE's style requirements, including those for file naming. The PLOS ONE style templates can be found at http://www.plosone.org/attachments/PLOSOne_formatting_sample_main_body.pdf and http://www.plosone.org/attachments/PLOSOne_formatting_sample_title_authors_affiliations.pdf**

Reviewers' comments:

Reviewer's Responses to Questions

**Comments to the Author**

1. Is the manuscript technically sound, and do the data support the conclusions?

Reviewer #1: Partly

Reviewer #2: Yes

2. Has the statistical analysis been performed appropriately and rigorously? 

Reviewer #1: I Don't Know

Reviewer #2: I Don't Know

3. Have the authors made all data underlying the findings in their manuscript fully available?

Reviewer #1: Yes

Reviewer #2: Yes

4. Is the manuscript presented in an intelligible fashion and written in standard English?

Reviewer #1: No

Reviewer #2: Yes

5. Review Comments to the Author

Reviewer #1: 1. This is an interesting paper in terms of its approach yet it strikes me as a preliminary step for a more comprehensive and detailed analysis of the SHARE Ware 1 and 6 data. Review of the original data suggest it would be feasible to perform analyses of social, economic, health, and mental health results that might account for differences in perceived chances of survival: this holds for gender differences, country differences, and temporal differences or similarities.

2. The Introduction could be condenses and better organized. Similarities and differences in published studies could be summarized more succinctly.

3. The data and methods section is more detailed than it needs to be and might be better as a Table.

4. I am a medical scientist and we would not employ terms like optimism or pessimism but frame our descriptions in terms of realistic or more accurate perceptions. There are data in Wave 1 and 6 survey that actually could be used to address "pessimism".

5. I find glibly excluding (the Danish data because it is difficult to account for in the model problematic. Does it point to problems with the model and its principles?

6. The main findings could be summarized in a more parsimonious fashion. That being said, there should be a more thoughtful discussion of:

a. the reasons for certain patterns of discrepancies between perception of life expectancy and actual life expectancy

b. the profound gender differences

c. what could account for some of the profound temporal data shifts such as in Austria, Germany, and Sweden.

Reviewer #2: Authors analysed data from the SHARE surveys Waves 1 and 6 in order to explore subjective life expectancy (LE) and compare with actual LE, based on Human Mortality Database. The paper has several strength, eg. analysis of several countries in parallel, comparing results of the same survey in two time points and comparing genders.

However, I think there are some weak points as well. In the SHARE database respondents are asked about probabilities, ie (Page6 Line 114): 'What are your chances that you will live to be age T or more?' Given the low statistical literacy level of the populations (please see studies by Gigerenzer G. et al) the responses are probably biased in a significant number of cases. Most people can often say the exact age they think they will reach if they are asked in asimple direct way (eg. https://www.ncbi.nlm.nih.gov/pubmed/26077549 ). Howveer, this question forces them to translate this idea first to a specific age (which is either over or below or equal to their subjctive LE). Then, in order to respond this question they have to express that in percentages. This methodology is my major concern regarding the whole study. I understand, of course, that the SHARE data is a given source and Authrs tend to make good use of it, however I am still doubtful about the validity and applicability of these responses to subjective LE estimates. I suggest, therefore, to test the validity of this question at least in a small sample.

My second concern is related to the comparison with actual LE. Sociodemographic status is a major determinant of LE. Was the SHARE sample matched to the controll sample (actual LE) by main socioeconomic characteristics, such as educational and income level? For instance, slight differences in educational level between the two may result in differences between subjective and actual LE without any real under- or overestimation. More educated people will express a longer subjective LE and in fact their actual LE is higher than of the average population, hence their subjective LE is realistic and the observed gap is artificial.

Overall, Authors do not provide enough infomation about the SHARE data sampling and characteristics of the sample which makes the interpretation of the results challenging for someone who is not very familiar with SHARE.

On Page7Line131 Author refer to 'limited abilities'. Is it the GALI question from the MEHM?

Analysis of determinants of over- and underestimation (regression) would increase substantially the scientific content of the study.

6. PLOS authors have the option to publish the peer review history of their article (what does this mean?). If published, this will include your full peer review and any attached files.

Reviewer #1: No

Reviewer #2: No

---

## [Author Response · Author response to Decision Letter 0]

6 Feb 2020

1. Is the manuscript technically sound, and do the data support the conclusions?

Reviewer #1: Partly

Reviewer #2: Yes

2. Has the statistical analysis been performed appropriately and rigorously? 

Reviewer #1: I Don't Know

Reviewer #2: I Don't Know

3. Have the authors made all data underlying the findings in their manuscript fully available?

Reviewer #1: Yes

Reviewer #2: Yes 

4. Is the manuscript presented in an intelligible fashion and written in standard English?

Reviewer #1: No

Reviewer #2: Yes

The submitted manuscript was language-edited. 

5. Review Comments to the Author

We thank the reviewers for their useful comments. Following the recommendations of the reviewers we introduced significant changes in the paper and considerably improved it. 

While applying the changes, we keep the focus of the paper on trends and observations of subjective life expectancies that have not been published elsewhere. 

The major changes include:

- The introduction is shortened.

- We entered a new table in the section on data and methods.

- We extended the Discussions. We elaborate more extensively the effect of important issues such as health and level of education. We also discuss non-representativeness of samples by education and find that it does not harm our inferences. 

- We excluded Denmark where non-representativeness by education is particularly large.

Reviewer #1: 1. This is an interesting paper in terms of its approach yet it strikes me as a preliminary step for a more comprehensive and detailed analysis of the SHARE Ware 1 and 6 data. Review of the original data suggest it would be feasible to perform analyses of social, economic, health, and mental health results that might account for differences in perceived chances of survival: this holds for gender differences, country differences, and temporal differences or similarities.

We agree with the reviewer that the paper is a preliminary step towards more comprehensive analyses. Our paper explores trends and observations related to subjective life expectancies that have not been published elsewhere. Explanations of such trends and observations can be research topics in subsequent papers. Yet in this paper we consider the effects of health and education, while rigorous analyses of these effects can be topics of separate papers. 

2. The Introduction could be condenses and better organized. Similarities and differences in published studies could be summarized more succinctly.

The introduction is by about half a page shorter which also improved the clarity.

3. The data and methods section is more detailed than it needs to be and might be better as a Table.

Done; Table 2 was added.

4. I am a medical scientist and we would not employ terms like optimism or pessimism but frame our descriptions in terms of realistic or more accurate perceptions. There are data in Wave 1 and 6 survey that actually could be used to address "pessimism".

These terms were replaced with others such as upward bias and downward bias. 

5. I find glibly excluding (the Danish data because it is difficult to account for in the model problematic. Does it point to problems with the model and its principles?

The Danish sample strongly over-represents people with higher education. The country is occupying exceptional positions in other international comparisons, such as for healthy life (for example in references 12, 13, 22). We excluded Denmark from our analysis as it cannot make a meaningful contribution. 

6. The main findings could be summarized in a more parsimonious fashion. That being said, there should be a more thoughtful discussion of:

a. the reasons for certain patterns of discrepancies between perception of life expectancy and actual life expectancy

b. the profound gender differences

c. what could account for some of the profound temporal data shifts such as in Austria, Germany, and Sweden.

With respect to points a. and b., similar issues have been extensively analyzed with respect to healthy and unhealthy life elsewhere. Given that there is no research for the same issues for subjective life expectancies, we decided to be careful and consider the issues in the Discussion. There, ee refer to the “male-female survival paradox” that seems similar to the gender difference which we found, and speculate that explanations could be correspondingly very similar. 

As for point c., we are restricted with respect to country-specific discussion which needs more detailed and thorough analyses of country-specific idiosyncrasy. 

Reviewer #2: Authors analysed data from the SHARE surveys Waves 1 and 6 in order to explore subjective life expectancy (LE) and compare with actual LE, based on Human Mortality Database. The paper has several strength, eg. analysis of several countries in parallel, comparing results of the same survey in two time points and comparing genders.

However, I think there are some weak points as well. In the SHARE database respondents are asked about probabilities, ie (Page6 Line 114): 'What are your chances that you will live to be age T or more?' Given the low statistical literacy level of the populations (please see studies by Gigerenzer G. et al) the responses are probably biased in a significant number of cases. Most people can often say the exact age they think they will reach if they are asked in asimple direct way (eg. https://www.ncbi.nlm.nih.gov/pubmed/26077549 ). Howveer, this question forces them to translate this idea first to a specific age (which is either over or below or equal to their subjctive LE). Then, in order to respond this question they have to express that in percentages. This methodology is my major concern regarding the whole study. I understand, of course, that the SHARE data is a given source and Authrs tend to make good use of it, however I am still doubtful about the validity and applicability of these responses to subjective LE estimates. I suggest, therefore, to test the validity of this question at least in a small sample.

This point is extremely important, and we have responded to it in the section on data and methods. The organizers have included a special question designed with the specific purpose to help respondents to evaluate chances of an event. The question is added in the text. For clarity we include it also here:

As this question may seem difficult to respondents, they were introduced to the issue of “chances” with an example. They were asked to answer the question: “What do you think are the chances that it will be sunny tomorrow?” with answers from 0 to 100. 

My second concern is related to the comparison with actual LE. Sociodemographic status is a major determinant of LE. Was the SHARE sample matched to the controll sample (actual LE) by main socioeconomic characteristics, such as educational and income level? For instance, slight differences in educational level between the two may result in differences between subjective and actual LE without any real under- or overestimation. More educated people will express a longer subjective LE and in fact their actual LE is higher than of the average population, hence their subjective LE is realistic and the observed gap is artificial.

Overall, Authors do not provide enough infomation about the SHARE data sampling and characteristics of the sample which makes the interpretation of the results challenging for someone who is not very familiar with SHARE.

Below is the paragraph from the discussion section where we address the issue:

“Our study is not free of caveats. Survey data are usually problematic with respect to how well they represent the population of study, in this case men and women aged 60 to 80 years. The SHARE data have been found to be non-representative regarding educational attainment: persons with higher education are over-represented in our list of countries (23). Due to this sample bias, estimated subjective life expectancies are higher than those of the study population. If samples were representative for education, subjective life expectancies would be lower and therefore our results would be reinforced, as downward bias would increase. An open question remains with respect to men’s realism in 2015; it could turn out that men were biased downwards in 2015 as well. To check the effect of non-representativeness by education we used crudely constructed weights by education for Austrian men in 2015 using their distribution by three levels of education (23,24). We found that the subjective life of these men would be a fraction of one year lower (i.e., 0.2) than the one displayed in Table 4a. We thus do not expect that non-representativeness by education harms our inferences. “

On Page7Line131 Author refer to 'limited abilities'. Is it the GALI question from the MEHM?

Analysis of determinants of over- and underestimation (regression) would increase substantially the scientific content of the study.

 It is indeed the GALI question. The question is cited in the section on data and methods. 

Analyses of determinants is crucially important, yet it can be a subject of a range of research topics and papers. In this paper we expose newly established main trends and observations on subjective life expectancies, and we hope they will attach attention among researchers.

6. PLOS authors have the option to publish the peer review history of their article (what does this mean?). If published, this will include your full peer review and any attached files.

Do you want your identity to be public for this peer review? For information about this choice, including consent withdrawal, please see our Privacy Policy.

Reviewer #1: No

Reviewer #2: No

---

## [Decision Letter · Decision Letter 1]

20 Feb 2020

Subjective Length of Life of European Individuals at Older Ages: Temporal and Gender Distinctions

PONE-D-19-30019R1

Dear Dr. Scherbov,

We are pleased to inform you that your manuscript has been judged scientifically suitable for publication and will be formally accepted for publication once it complies with all outstanding technical requirements.

With kind regards,

Gianluigi Forloni

Academic Editor

PLOS ONE

Additional Editor Comments (optional):

Reviewers' comments:

Reviewer's Responses to Questions

**Comments to the Author**

1. If the authors have adequately addressed your comments raised in a previous round of review and you feel that this manuscript is now acceptable for publication, you may indicate that here to bypass the “Comments to the Author” section, enter your conflict of interest statement in the “Confidential to Editor” section, and submit your "Accept" recommendation.

Reviewer #1: All comments have been addressed

2. Is the manuscript technically sound, and do the data support the conclusions?

Reviewer #1: Yes

3. Has the statistical analysis been performed appropriately and rigorously? 

Reviewer #1: Yes

4. Have the authors made all data underlying the findings in their manuscript fully available?

Reviewer #1: Yes

5. Is the manuscript presented in an intelligible fashion and written in standard English?

Reviewer #1: Yes

6. Review Comments to the Author

Reviewer #1: The authors did a reasonable job being responsive to the review. I do worry that excluding Denmark is a form of cherry picking.

7. PLOS authors have the option to publish the peer review history of their article (what does this mean?). If published, this will include your full peer review and any attached files.

Reviewer #1: No

---

## [Editor Report · Acceptance letter]

27 Feb 2020

PONE-D-19-30019R1 

Subjective Length of Life of European Individuals at Older Ages: Temporal and Gender Distinctions 

Dear Dr. Scherbov:

I am pleased to inform you that your manuscript has been deemed suitable for publication in PLOS ONE. Congratulations! Your manuscript is now with our production department. 

With kind regards,

on behalf of

Dr. Gianluigi Forloni 

Academic Editor

PLOS ONE